# M³E: A Unified Framework for Large-Scale Multimodal Embedding via Multi-Task Mixture-of-Experts

## Abstract

Universal multimodal embeddings are crucial for enabling downstream tasks such as cross-modal retrieval and retrieval-augmented generation. With the powerful semantic understanding capabilities of Large Vision-Language Models (LVLMs), leveraging them for embedding learning has emerged as a new paradigm. Recent research has primarily focused on prompt engineering or synthesizing high-quality training samples to enhance embedding quality. Although significant progress has been made, these methods often overlook the task diversity inherent in general-purpose embedding learning. This leads to two major issues: (1) The presence of too many easy or false negative samples degrades the discriminative power of the learned representations; (2) Diverse training tasks can lead to task conflict and oblivion problems. In this paper, we propose a unified multimodal multi-task embedding framework M³E that integrates innovations at both the data and model levels. On the data side, we utilize a Hard Negative-Aware Sample Scheduler (HNASS) module to increase the proportion of hard negative samples. In addition, to reduce easy negatives sample in the batch, we ensure the samples in a batch come from the same task dataset. While optimization for different tasks should be decoupled to avoid task conflicts. So on the model side, we design a Task-wise Low-Rank Mixture of Experts (Task-wise MOE) module that allocates task-specific experts to capture specialized representations, while shared experts are used to learn generalizable cross-task knowledge. This effectively mitigates inter-task conflicts and improves the stability of multi-task learning. Extensive experiments demonstrate that our method significantly improves the embedding performance of LVLMs across 36 tasks. Our code will be released.

## 1 Introduction

Multimodal embedding encodes multimedia inputs into latent vector representations, providing general support for a range of downstream tasks such as semantic matching (Li et al., 2024b; Wang et al., 2024b), information retrieval (Li et al., 2025; Dai et al., 2024a), in-context learning (Suo et al., 2024; Li et al., 2024a), and retrieval-augmented generation (RAG) (Yu et al., 2024; Xiong et al., 2024). Previous studies (e.g., CLIP (Radford et al., 2021)) employ separate encoders for image and text modalities, which inherently lacks robust interaction mechanisms between different modalities. This inevitably limited the model's effectiveness in multimodal tasks requiring complex reasoning. With the development of Large Vision-Language Models (LVLMs) (Bai et al., 2023; Dai et al., 2024b), recent research has increasingly focused on utilizing LVLMs as unified embedding frameworks. By integrating semantic understanding with cross-modal alignment capabilities, These approaches present a promising direction for constructing robust and universal embeddings.

To improve the embedding capabilities of LVLMs, existing approaches can be roughly categorized into two research lines. As shown in Fig. 1 (a), the first strategy (Zhang et al., 2024; Jiang et al., 2024a) operates at the model level, employing task-specific prompts to enhance the embedding performance of LVLMs across different tasks; Although different task prompts are used, the fact that all tasks share a set of parameters still leads to gradient conflicts and task forgetting problems during training. Conversely, as shown in Fig. 1 (b), the second strategy (Chen et al., 2025; Zhou et al., 2024; Thirukovalluru et al., 2025) focuses on the data level, where high-quality contrastive samples

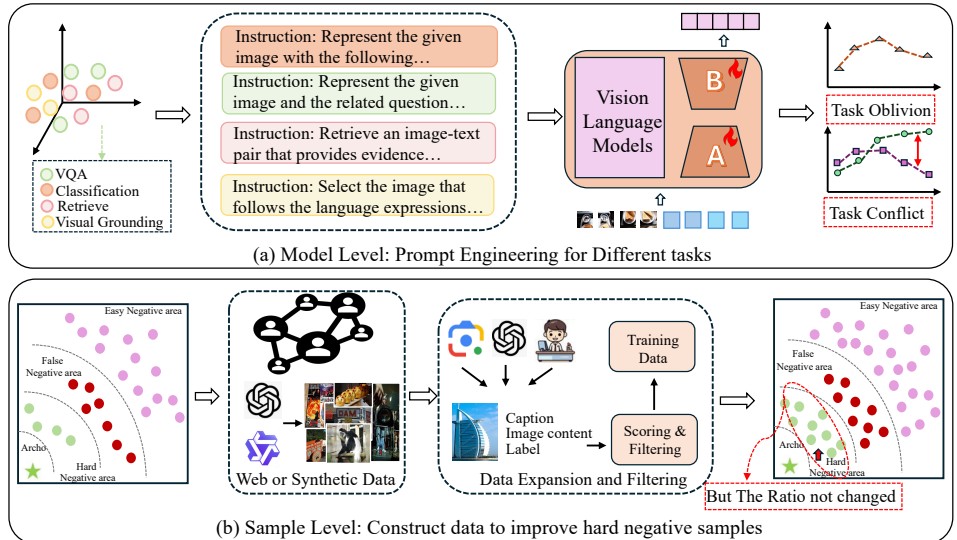

Figure 1: Challenges associated with existing embedding methods. (a) Model Level: Using task-specific prompts to enhance the representation capabilities of LVLMs across different tasks, while facing challenges such as gradient conflicts and task oblivion. (b) Sample Level: Constructing high-quality contrastive samples to boost hard negatives samples in contrastive learning, but still has excessive uninformative samples (easy or false negatives) since the proportion of negative types stays unchanged.

are constructed to improve the number of negative samples in contrastive learning. Although this approach increases the number of hard negatives, the relative proportion of easy, false, and hard negatives remains unchanged, and random sampling still yields an excessive number of uninformative (i.e., easy or false negative) samples.

In summary, the above methods do not consider the diversity of tasks, which leads to two key challenges: 1) **Difficulty in Negative Sample Selection:** Due to the ambiguous semantic boundaries between tasks, the training process is prone to introducing false negatives or easy negatives, which weakens the discriminative power of contrastive learning. 2) **Task Conflict and Oblivion**: Simultaneously training multiple tasks may lead to optimization conflicts, where the learning objectives of one task interfere with others. In addition, the introduction of new tasks may deteriorate the performance of previously learned tasks.

To tackle the above challenges, we exploit a unified **M**ulti**M**odal **M**ulti-task **E**mbedding framework M$^3$E that simultaneously tackles issues at both the data and model levels. Specifically, at the data level, we propose the Hard Negative-Aware Sample Scheduler(HNASS) for mining challenging negatives. We first construct a sample similarity graph using teacher model and apply graph clustering algorithm to partition training samples. This groups semantically similar samples together while separating dissimilar ones, thereby increasing the number of hard negative samples in each batch and enhancing the training stability and effectiveness of contrastive learning. In addition, since the presence of multi-task samples in a batch will lead to an excessive number of easy negative samples, we build a separate graph for each dataset to ensure that the samples in each batch belong to the same task. At the same time, to alleviate the problems of gradient conflict and forgetting, the optimization of different tasks should only focus on a small number of parameters, the parameter space should be decoupled from each other. Therefore, at the model level, we introduce the Task-wise Low Rank Mixture of Experts (Task-wise MOE) structure, which consists of task experts and shared experts. The task experts captures task-specific representations, while the shared experts learns general knowledge across tasks. To achieve flexible task adaptation, we dynamically adjust the weights of task experts and shared experts through a router. This design not only effectively alleviates task conflicts, but also mitigates the issue of catastrophic forgetting to some extent. In summary, we make the following contributions:

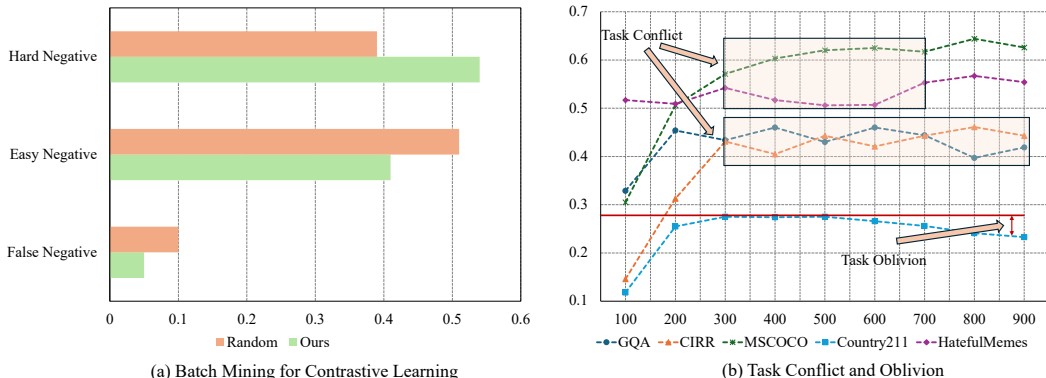

Figure 2: (a) Proportions of easy negatives, false negatives, and hard negatives in batches constructed with different sampling methods. (b) Performance evolution across five datasets covering different task types during training.

1) Our work reveals the challenges of universal embeddings arising from multi-task learning and proposes a unified framework $M^3E$ to mitigate these challenges from both the data and model levels.

2) We propose a low-rank mixture of experts, comprising both task experts and shared experts, to mitigate the problems of task conflict and oblivion.

3) Our model achieves state-of-the-art performance across 4 tasks and 36 datasets, while also demonstrating excellent generalization.

## 2 RELATED WORK

### 2.1 MASSIVE MULTIMODAL EMBEDDING

In recent years, representation learning for text and images has become an important research direction. With the development of Large Vision-Language Models (LVLMs), recent studies have started to build embedding representations based on LVLMs. Current studies mainly focuses on two directions: Firstly, at the data level, the emphasis is on constructing high-quality data to enhance representation capabilities (Chen et al., 2025; Zhou et al., 2024; Thirukovalluru et al., 2025). For example, mmE5 (Chen et al., 2025) proposed a data synthesis framework that leverages MLLM to generate diverse task, modality, and language combinations, thereby improving the generalization and robustness of the embeddings. Secondly, at the model level, the focus is on prompt design, using task-specific prompts to enhance embedding performance in specific scenarios (Zhang et al., 2024; Jiang et al., 2024a) . For example, E5-V (Jiang et al., 2024a) effectively reduces the modality gap between different input types through task-specific prompts.

### 2.2 HARD NEGATIVE SAMPLE MINING

Several studies (Sachidananda et al., 2023; Yang et al., 2023; Thirukovalluru et al., 2025; Kim et al., 2025) have explored improving contrastive learning model performance by optimizing the batch construction process. For example, BatchSampler (Yang et al., 2023) uses random walks on an adjacency graph to sample batches. B3++ (Thirukovalluru et al., 2025) constructs batches using graph clustering algorithms and selects common hard negative samples for each batch. FALCON (Kim et al., 2025) proposes a reinforcement learning-based mini-batch construction strategy to balance the trade-off between hard negatives and false negatives. However, these methods do not fully consider the negative impact of excessive hard negative samples on batch construction, and their complexity is high when applied to large-scale datasets. More related work can be found in Appendix A.1.

## 3 PRELIMINARY

### 3.1 LARGE VISION-LANGUAGE MODELS EMBEDDING PIPELINE

Given a Large Vision-Language Models (LVLMs) with parameters $\theta$, the model can effectively perform multi-modal embedding tasks by taking the last layer vector representation of the last token, then InfoNCE loss is used to train LVLMs. Specifically, for a given input query $q$ and target $t$,:

$$\mathcal{L} = -\log \frac{f_k(\mathbf{h}_q, \mathbf{h}_{t+})}{f_k(\mathbf{h}_q, \mathbf{h}_{t+}) + \sum_{t^- \in \mathcal{N}} f_k(\mathbf{h}_q, \mathbf{h}_{t-})} \tag{1}$$

Where $h_q$ represents the embedding of the query, $h_{t+}$ represents the embedding of the target, $\mathcal{N}$ represents the remaining samples in the batch, which are treated as negative samples in contrastive learning. $f_k$ serves as a function to compute the matching score between query embedding $h_q$ and target embedding $h_t$. Following (Jiang et al., 2024b), the temperature-scaled cosine similarity is adopted as the concrete form of $f_k$, which is formulated as $f_k(h_q, h_t) = exp(\frac{1}{\tau}cos(h_q, h_t))$, where $\tau$ denotes a temperature hyper-parameter.

In recent years, the focus of research has gradually shifted towards building universal embedding representations, which can generalize well across tasks and domains. While the diversity of task types introduces issues at both the data and model levels. Firstly, due to the variety of task types, randomly constructed batches are prone to introducing a high number of false ans easy negative samples. Secondly, multi-task training can lead to task conflicts and oblivion issues: i) training multiple tasks simultaneously may cause the optimization direction of one task to interfere with others, affecting the model's generalization ability; ii) in continual learning scenarios, learning new tasks may degrade the performance of previously learned tasks. Next, we will conduct related experiments to investigate these issues.

### 3.2 BATCH MINING FOR CONTRASTIVE LEARNING

To analyze the composition of negative samples, we randomly select 20,480 instances from the MMEB-V1 dataset (Jiang et al., 2024b) and divide them into mini-batches using both random sampling and our proposed HNASS method (detailed in Sec. 4.2). A trained embedding model is employed to generate representations for each sample, serving as a discriminator to estimate pairwise similarity. Based on predefined thresholds $\delta$ and $\eta$, sample pairs within each batch are categorized as follows: those with similarity above $\eta$ are considered false negatives; those below $\delta$ are treated as easy negatives; and the remaining are regarded as hard negatives. Specifically, easy negative samples refer to samples that are easily distinguishable during training and provide little to no benefit to the model's learning process. False negative samples are positive samples that are incorrectly identified as negative during training. Hard negative samples are those that are difficult for the model to distinguish from positive samples, and they play a crucial role in improving the model's performance by providing more challenging learning signals. The distribution of these categories across batches is visualized in Fig. 2 (a).

We observe that batches generated via random sampling contain an overwhelming proportion of easy negatives, which hampers learning by providing weak discriminative information during training. In contrast, our HNASS method substantially increases the presence of hard negatives, effectively filtering out false negatives and introducing informative, moderately challenging examples. This leads to a more discriminative embedding space and provides a stronger foundation for downstream training.

### 3.3 TASK CONFLICT AND OBLIVION

To investigate task conflict and oblivion in multi-task training, we select five datasets spanning diverse task types: classification (Country211, HatefulMemes), visual question answering (GQA), visual grounding (MSCOCO), and image-text retrieval (CIRR). They form a heterogeneous task suite for evaluating model behavior under joint training. During training, all tasks are learned simultaneously using uniform hyperparameters, and performance dynamics for each task are recorded throughout the process.

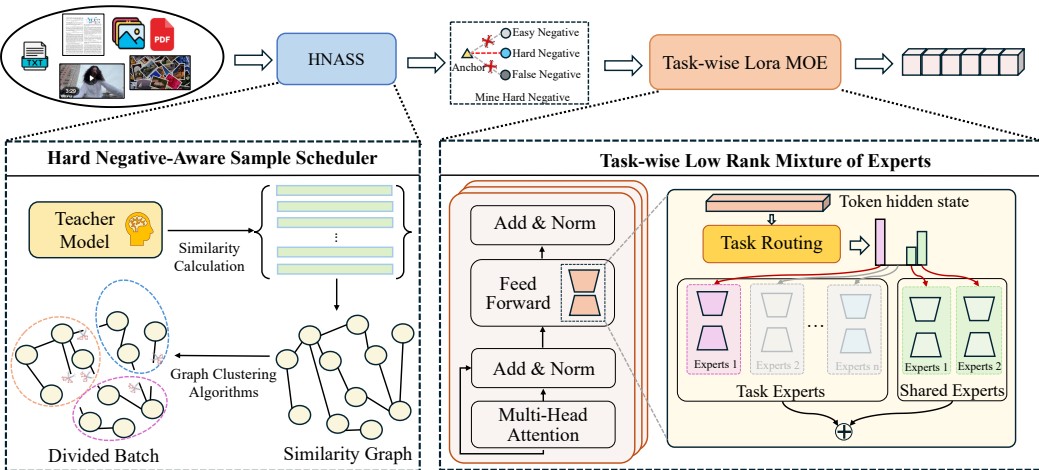

Figure 3: **Overview of our** M³E. Our framework consist of two key compontents: Hard Negative-Aware Sample Scheduler (HNASS) and Task-wise Low Rank Mixture of Experts (Task-wise MOE). Specifically, we first construct high-quality hard negative batches using HNASS to improve the stability and effectiveness of training. Then, the constructed batches are fed into TLMoe. Different tasks are embedded using dedicated task experts and shared experts.

As illustrated in Fig. 2 (b), we observe a clear trade-off between performance on the GQA and CIRR tasks: improvements in one often correspond to declines in the other. A similar phenomenon emerges in the joint training of MSCOCO and HatefulMemes datasets, suggesting the presence of parameter competition or representation interference between tasks. Additionally, the performance on Country211 consistently deteriorates throughout training, reflecting a pronounced task oblivion issue, which means that the model gradually loses its ability to represent the task when focusing on other tasks. These observations show that when multiple tasks share the same model parameters, conflicts and forgetting between tasks will significantly affect the overall training effect, making it difficult for the model to achieve optimal performance on all tasks at the same time.

### 3.4 DISCUSSION

Based on the above experiments, it can be found that the diversity of task types introduces challenges at both the data and model levels. On the one hand, randomly constructed batches contain a large number of easy negatives, which leads to inefficient learning. On the other hand, task conflict and forgetting caused by multi-task learning severely hinder model performance, making it difficult to achieve a balanced optimization across tasks. Prior approaches often address data-level and model-level issues in isolation, lacking a unified optimization framework. In light of this, we propose a unified multimodal embedding framework M³E that tackles multi-task challenges holistically from both the data and model perspectives, aiming to improve the overall quality of the learned embeddings.

## 4 OUR METHOD

### 4.1 MODEL OVERVIEW

Based on the discussion in Sec. 3, we design a new unified multi-task embedding framework $\mathbf{S^3C}$, which aims to uniformly address the problems brought by multi-task learning from both the data and model levels. As shown in Fig. 3, Our framework comprises an Hard Negative-Aware Sample Scheduler (HNASS) and Task-wise Low Rank Mixture of Experts (Task-wise MOE). Firstly, for a multi-task dataset, we first construct high-quality hard negative batches using HNASS to improve the stability and effectiveness of training. Then, the constructed batches are fed into the Task-wise Low Rank Mixture of Experts. Different tasks are embedded using dedicated task experts and shared experts. The task experts decouple different tasks in the parameter space, while the shared experts

learn the common knowledge between different tasks, thereby alleviating conflicts between tasks and improving the stability and performance of multi-task learning. Next, we will provide a detailed introduction to the Hard Negative-Aware Sample Scheduler and the Task-wise Low Rank Mixture of Experts methods.

## 4.2 HARD NEGATIVE-AWARE SAMPLE SCHEDULER

As analyzed in Sec. 3.2, randomly constructed batches contain a large number of easy negative samples, which hinders the model's ability to effectively learn discriminative embeddings. In addition, the presence of false negative samples within the batch will lead the optimization process in an incorrect direction.

To address these issues, we first adopt a pretrained model as the teacher model to compute embeddings for all training samples. Using these embeddings, we construct a similarity matrix $S$, where each element $S_{ij}$ denotes the similarity score between sample $i$ and sample $j$. In order to eliminate easy negatives and filter out false negatives, we exclude the top-$p$ most similar samples for each instance and retain the bottom-$m$ ranked ones as hard negatives for batch mining. Specifically, we construct a sparse graph using the submatrix $S_f = S[:, p : p + m]$, which serves as the adjacency list of the graph. In this graph, nodes represent individual samples, and edge weights correspond to their similarity scores. Then following (Thirukovalluru et al., 2025), we employ the METIS (Liu et al., 2015) community clustering algorithm to partition the graph. METIS is a minimum-cut based algorithm that maximizes intra-cluster edge density while minimizing inter-cluster connections. Through this method, we identify clusters of size $c$. Given a target batchsize B, we randomly sample $B/C$ clusters to construct a new training batch.

By using METIS for graph partitioning, we can ensure that samples within each subgraph have a high degree of similarity. Because false negative samples have been filtered out, the samples in each subgraph after partitioning are essentially hard negative samples, which are difficult to distinguish. This partitioning method not only effectively increases the challenge of model training but also promotes effective learning of difficult negative samples, thereby improving the optimization of contrast loss during training. The complete procedure is illustrated in Appendix A.2.

Unlike the B3++ method, the main differences in our approach are: i) B3++ selects all common hard negative samples for each batch, which results in higher time complexity. ii) During training, we iteratively update the teacher model using the trained model to improve the initial embedding performance.

## 4.3 TASK-WISE LOW RANK MIXTURE OF EXPERTS

Since the presence of multi-task samples within a batch can lead to an excess of easy negative samples, in the Sec. 4.2 we constructed separate graphs for each dataset to ensure that all samples within a batch belong to the same task. Additionally, in order to mitigate the task conflicts and forgetting issues discussed in Sec. 3.3, the optimization of different tasks should focus only on a small subset of parameters, with the parameter spaces decoupled from one another.

To effectively decouple the entanglement of different tasks in the parameter space, we propose the Task-wise Low-Rank Mixture of Experts module, which integrates multiple low-rank adapters with a task-aware routing mechanism. This design enhances the stability and representational capacity of multi-task learning. Specifically, when fine-tuning large vision-language models (LVLMs) with LoRA, instead of reconstructing each fully connected layer with a single pair of low-rank matrices, we introduce multiple LoRA experts for each layer. Each expert consists of a pair of trainable matrices $\mathbf{A}_i \in \mathbb{R}^{d_q \times r}$ and $\mathbf{B}_i \in \mathbb{R}^{r \times d_p}$, where $r \ll \min(d_q, d_p)$. These matrices are initialized with a Gaussian distribution and zeros, respectively.

For a model with $m$ Transformer layers, we assign $N$ experts to each layer, which are further divided into a task-specific expert set $E_t$ and a shared expert set $E_s$, responsible for modeling task-specific and task-invariant knowledge, respectively. Meanwhile, we introduce a router $S$ with learnable weights $\mathbf{W} \in \mathbb{R}^{d_q \times N}$, which dynamically selects an appropriate combination of experts based on the input token $x$. The updated weights are then obtained by combining the pretrained parameters with the weighted outputs of multiple experts, formulated as:

Table 1: Comparison with the state-of-the-art methods for the MMEB dataset across four different tasks. Specifically, our method improves by 10.3 and 7.8 points compared to the baseline Qwen2-VL(2B) and Qwen2-VL(7B). The underline represents the second highest result.

| Model | Retrieval | Classification | VQA | Viual Grounding | Avgrage |
|---|---|---|---|---|---|
| #Tasks | 12 | 10 | 10 | 4 | 36 |
| *LIP Style Baselines* | | | | | |
| CLIP (Radford et al., 2021) | 42.8 | 9.1 | 53.0 | 51.8 | 37.8 |
| BLIP2 (Li et al., 2023) | 27.0 | 4.2 | 33.9 | 47.0 | 25.2 |
| SigLIP (Zhai et al., 2023) | 40.3 | 8.4 | 31.6 | 59.5 | 34.8 |
| OpenCLIP (Cherti et al., 2023) | 47.8 | 10.9 | 52.3 | 53.3 | 39.7 |
| UniIR (BLIP$_{FF}$) (Wei et al., 2024) | 42.1 | 15.0 | 60.1 | 62.2 | 42.8 |
| UniIR (CLIP$_{SF}$) (Wei et al., 2024) | 44.3 | 16.2 | 61.8 | 65.3 | 44.7 |
| MagicLens (Zhang et al., 2024) | 38.8 | 8.3 | 35.4 | 26.0 | 27.8 |
| *∼2B VLM Models (Trained on MMEB)* | | | | | |
| VLM2Vec (Qwen2-VL-2B) (Jiang et al., 2024b) | 65.4 | 59.0 | 49.4 | 73.4 | 59.3 |
| UniME (Phi-3.5-V) (Gu et al., 2025) | 64.5 | 54.8 | 55.9 | 81.8 | 64.2 |
| LLaVE (Aquila-VL-2B) (Lan et al., 2025) | 65.2 | 62.1 | 60.2 | 84.9 | 65.2 |
| B3 (InternVL3-2B) (Thirukovalluru et al., 2025) | 69.0 | 62.6 | 64.0 | **86.9** | 67.8 |
| B3++ (Qwen2-VL-2B) (Thirukovalluru et al., 2025) | 70.9 | **67.0** | 61.2 | 79.9 | 68.1 |
| **M³E (Qwen2-VL-2B)** | **72.1** | 65.9 | **64.8** | 86.4 | **69.9** |
| *>7B VLM Models (Trained on MMEB)* | | | | | |
| VLM2Vec (Qwen2-VL-7B) (Jiang et al., 2024b) | 69.9 | 62.6 | 57.8 | 81.7 | 65.8 |
| MMRet (Llava-Next-7B) (Zhou et al., 2024) | 69.9 | 56.0 | 57.4 | 83.6 | 64.1 |
| mmE5 (Llama-3.2-11B) (Chen et al., 2025) | 70.9 | 67.6 | 62.8 | 89.7 | 69.8 |
| LLaVE (Llava-OV-7B) (Lan et al., 2025) | 70.9 | 65.7 | 65.4 | **91.9** | 70.3 |
| UniME (LLaVA-OneVision-7B) (Gu et al., 2025) | 70.5 | 66.8 | 66.6 | 90.9 | 70.7 |
| B3 (InternVL3-7B) (Thirukovalluru et al., 2025) | 73.2 | 65.0 | 68.8 | 91.8 | 71.8 |
| B3++ (Qwen2-VL-7B) (Thirukovalluru et al., 2025) | 74.1 | **70.0** | 66.5 | 84.6 | 72.0 |
| **M³E (Qwen2-VL-7B)** | **75.5** | 68.8 | **69.5** | 90.0 | **73.6** |

$$\mathbf{O} = \mathbf{W}x + \Delta\mathbf{W}(x)x,$$

$$\Delta\mathbf{W}(x) = \sum_{k=1}^{N} \alpha_k(x) \cdot \mathbf{A}_k \mathbf{B}_k$$

$$\alpha_k(x) = \text{Softmax}(\text{Select}(S(x)))$$

where $\alpha_k(x)$ denotes the routing weight computed by the router for expert $k$ given the input $x$. The method Select for choosing task experts can be implemented using either a Top-k or a Mask-based approach. The Mask-based method selects specific experts based on the task type, while the Top-k strategy picks the top $k$ experts with the routing scores. This mechanism not only alleviates task interference but also improves the generalization ability of the model in multi-task scenarios.

## 5 EXPERIMENT

### 5.1 EXPERIMENTAL SETTING

**Training Datasets.** We conducted experiments on MMEB-V1 dataset (Jiang et al., 2024b), which comprises four types of tasks: visual question answering, classification, retrieval, and visual grounding. All tasks are reformulated as ranking problems, where the model is given an instruction and a query, and the goal is to select the most appropriate answer from a set of candidates.

**Evaluation Datasets.** We conduct a systematic evaluation of our model on the MMEB-V1 benchmark, which includes 36 tasks across four different tasks. Following (Meng et al., 2025), for all tasks, we use Precision@1 as the primary metric to measure the proportion of queries where the correct target is ranked first.

**Implementation Details.** We train our model using Qwen2-VL (Wang et al., 2024a) with both 2B and 7B as backbone and a batch size of 1024. We apply LoRA tuning with a rank of 8 and a scaling factor $\alpha = 32$, training for 2000 steps. All models are trained on 16 A100 GPUs. We adopt a learning rate of $1 \times 10^{-4}$ with a 10% warm-up ratio. For Hard Negative-Aware Sample Scheduler,

Table 2: **Ablation study.** "HNASS" represents the Hard Negative-Aware Sample Scheduler for mining challenging negatives. "Task-wise MOE" represents the Task-wise Low Rank Mixture of Experts module.

| HNASS | Task-wise MOE | Ret. | Cla. | VQA | Gro. | ID | OOD | Avg. |
|---|---|---|---|---|---|---|---|---|
| | | 65.4 | 59.0 | 49.4 | 73.4 | 66.0 | 52.6 | 59.3 |
| ✓ | | 70.2 | 61.0 | 54.1 | 82.4 | 70.1 | 55.1 | 64.5 |
| | ✓ | 67.1 | 58.5 | 55.9 | 80.1 | 69.6 | 51.9 | 63.1 |
| ✓ | ✓ | 72.1 | 65.9 | 64.8 | 86.4 | 73.8 | 63.1 | 69.9 |

Table 3: The effects of different Lora MOE architectures. More detailed discussion can be found in Sec. 5.3.

| Model | Ret. | Cla. | VQA | Gro. | ID | OOD | Avg. |
|---|---|---|---|---|---|---|---|
| *Backbone: Qwen2-2B;* | *Batch Size ($|B|$): 1024* | | | | | | |
| #Task : #Shared (1:2) | 72.1 | 65.9 | 64.8 | 86.4 | 73.8 | 63.1 | 69.9 |
| #Task : #Shared (2:1) | 71.8 | 64.7 | 65.0 | 84.8 | 74.0 | 61.6 | 69.4 |
| Top-k strategy | 71.4 | 63.8 | 64.4 | 83.5 | 73.7 | 60.7 | 68.7 |
| Mask-based strategy | 72.1 | 65.9 | 64.8 | 86.4 | 73.8 | 63.1 | 69.9 |

the hyperparameters $m$, $p$ follow the configuration in B3++ (Thirukovalluru et al., 2025). More details of evaluation metrics can be found in Appendix A.3.

## 5.2 QUANTITATIVE EVALUATION

In Table 1, we present a performance comparison on the MMEB dataset across four different tasks, including classification, visual question answering, visual grounding, and retrieval. Two multimodal large model baselines of different sizes are applied to evaluate the results, including QWen2-VL 2B and 7B. In particular, we observe that our method significantly improves the average performance across 36 datasets compared to previous methods. Furthermore, compared to the original baselines, our approach achieves an increase of 10.3 points on the 2B model and 7.8 points on the 7B model.

These results demonstrate the effectiveness of our approach. Our method performs exceptionally well on retrieval and visual grounding, which are key applications of embedding models. Previous methods did not account for the challenges introduced by multi-task learning in the process of general-purpose embedding. In contrast, we propose a universal embedding framework that addresses the challenges posed by multi-task learning at both the data and model levels, significantly improving the performance of each subtask. More detailed experimental results can be found in Appendix A.4.

## 5.3 ABLATION STUDIES

**Model Components Ablation.** In this section, we conduct several ablation studies on the MMEB-V1 dataset to systematically evaluate the contributions of different model components. As shown in rows 1-2 of Table 2, we can observe that the batch constructed with HNASS outperforms the baseline model by approximately 8.8%, confirming the effectiveness of HNASS in improving the quality of training samples. The results from the third row indicate that introducing Task-wise MOE leads to consistent performance improvements across various sub-tasks. This suggests that the mechanism effectively alleviates common issues in multi-task learning. Furthermore, the results in the last row show that combining HNASS with Task-MoE yields a 17.9% improvement over the baseline. This further confirms the complementary and synergistic relationship between the two modules, which together address multi-task learning challenges from both the data and model levels.

**MOE Architectures Ablation.** In Table 3, we exploit some alternative experiments about the MOE Architectures settings on the MMEB-V1dataset. Specifically, we first investigate the impact of the ratio between task-specific experts and shared experts, configuring them with 1:2 and 2:1 propor-

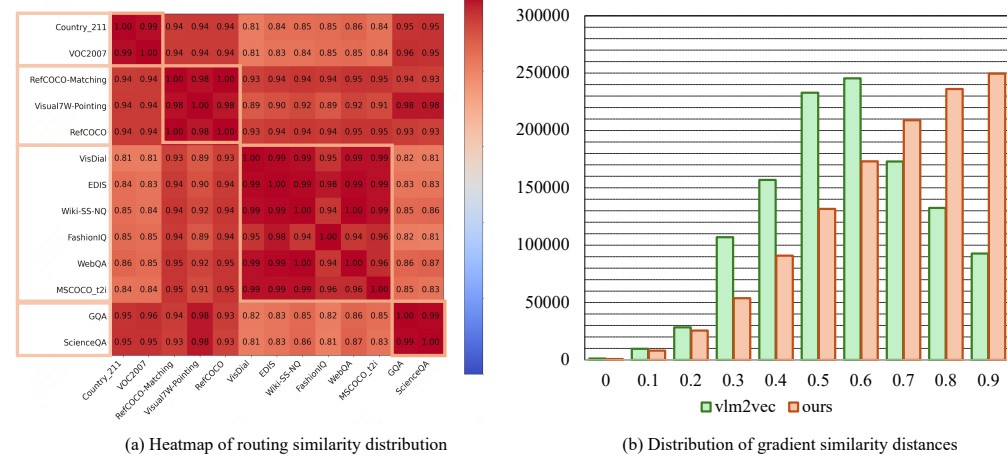

(a) Heatmap of routing similarity distribution      (b) Distribution of gradient similarity distances

Figure 4: (a) Heatmap of routing similarity distribution for different tasks. (b) Distribution of gradient similarity distances for training classification and visual grounding datasets.

tions respectively. The results from Rows 1 and 2 indicate that the performance difference between the two settings is marginal, suggesting that assigning one expert per task is generally sufficient. Subsequently, in Rows 3 and 4, we examine the effect of different expert selection strategies. The results show that the Top-k strategy performs worse than the Mask-based strategy. We attribute this performance gap to potential errors introduced by the top-k mechanism, while the Mask-based approach successfully avoids. More experimental can be found in Appendix A.5.

## 5.4 IN-DEPTH DISSECTION OF OUR MODEL

***Q1: How are routing distributions among different tasks?*** To investigate whether the router in Task-MoE is capable of learning task-specific characteristics, we visualize the similarity heatmap of routing distributions across tasks from different datasets, as shown in Fig. 4. The visualization reveals that datasets belonging to the same task tend to exhibit more consistent routing patterns. In contrast, tasks with greater differences, especially those from distinct categories, display more divergent routing behaviors. These observations suggest that Task-MoE can effectively capture both the commonalities and differences among tasks, thereby enabling more reasonable routing decisions in multi-task scenarios.

***Q2: Does Task-MoE alleviate the issue of gradient conflicts from different tasks?*** Due to the use of task-specific experts for different tasks, Task-MoE naturally decouples the parameter spaces between different tasks. As shown in Fig. 4, we visualized the cosine distance distribution of training gradients between the classification task and the visual grounding task. The results indicate that our method significantly reduces gradient conflicts for the parameters shared across tasks.

## 6 CONCLUSION

In this paper, we first explore the challenges that multi-task learning brings to general-purpose embedding models. Extensive experiments demonstrate that multi-task learning leads to difficulties in hard negative sample construction, as well as issues with task conflicts and forgetting. Based on these insights, we propose a universal multimodal embedding framework $M^3E$ that addresses the challenges of multi-task learning from both the data and model perspectives. Specifically, at the data level, we introduce the Hard Negative-Aware Sample Scheduler to mine challenging negative samples; at the model level, we propose the Task-wise Low-Rank Mixture of Experts structure to alleviate task conflicts and forgetting, enhancing the stability and performance of multi-task learning. More importantly, our method demonstrates excellent generalization performance across 36 datasets. We hope that this work will provide a universal framework for general-purpose multimodal embedding.

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

# A APPENDIX

## A.1 RELATED WORK: MULTI-TASK LEARNING

Multi-task learning (MTL) aims to enhance the overall performance of models by simultaneously learning multiple downstream tasks. Some studies (Javaloy & Valera, 2021; Chen et al., 2020) balance the loss or gradients of different tasks to avoid conflicts between them, or decouple shared information from task-specific information through manual design or automated architecture learning . With the development of Mixture of Experts (MoE) (Jacobs et al., 1991) systems, many studies (Zhou et al., 2025; Dimitriadis et al., 2024; Feng et al., 2025) have begun leveraging MoE for multi-task learning. For example, RecFound (Zhou et al., 2025) uses MoE to handle both embedding and generation tasks, while PaLoRA (Dimitriadis et al., 2024) introduces task-specific low-level adapters to improve the parameter efficiency of neural networks.

## A.2 HARD NEGATIVE-AWARE SAMPLE SCHEDULER ALGORITHMS

---
**Algorithm 1** Hard Negative-Aware Sample Scheduler

---
**Input**: Dataset $\mathcal{D}$, batchsize $B$, hyperparameters $p, m$.
1: Compute embeddings $\mathbf{e}_i$ for all samples in $\mathcal{D}$ using a pretrained teacher model;
2: Construct similarity matrix $S \in \mathbb{R}^{|\mathcal{D}| \times |\mathcal{D}|}$ where $S_{ij}$ is the similarity between $\mathbf{e}_i$ and $\mathbf{e}_j$;
3: Form submatrix $S_f = S[:, p : p + m]$ by excluding top-$p$ similar samples and retaining bottom-$m$ hard negatives for each instance.;
4: Build sparse graph $\mathcal{G}$ with nodes as samples and edge weights from $S_f$.;
5: Partition $\mathcal{G}$ into clusters $\mathcal{C} = \{C_1, \ldots, C_K\}$ using METIS ;
6: Randomly sample $k = \lfloor B/c \rfloor$ clusters from $\mathcal{C}$;
7: Collect all samples from the $k$ clusters to form $\mathcal{B}$;
8: **return** $\mathcal{B}$

---

## A.3 FURTHER IMPLEMENTATION DETAILS

For the 20 training datasets, if a dataset contains more than 100K samples, we sequentially select the first 100K to maintain consistency across training runs. When using GradCache, the sub-batch size is set to 2, and the total batch size is accumulated to 1,024. The initial Teacher model is based on VLM2Vec, and the subsequently trained models are iteratively reused as new Teacher models to generate updated batches. The temperature used was 0.02.

## A.4 DETAILED RESULTS OF ALL DATASETS

In Table 6, we present the detailed experimental results of several baselines and our $M^3E$ on MMEB-V1, which includes 20 in-distribution datasets and 16 out-of-distribution datasets. The 16 out-of-distribution datasets are highlighted with a yellow background in the table.

## A.5 ADDITIONAL EXPERIMENTS

In this section, we present additional experiments to further validate the effectiveness of our proposed method. In Sec A.5.1, we demonstrate the scalability of our approach by replacing different Large Vision Language Models (LVLMs), highlighting the generalizability of our method across various model architectures. In Sec A.5.2, we investigate Task-wise MOE architectural designs, exploring how varying the number of task experts at each layer impacts performance. Lastly, in Sec A.5.3, we study the impact of different ratios of hard negative samples on model performance.

### A.5.1 DIFFERENT LVLMS BACKBONE

In Table 4, we explore the impact of different Large Vision-Language Models backbones on the performance of our model. From the results in the first and third rows, it is evident that compared to Qwen-VL, InternVL3 performs better on the Visual Grounding task, likely due to its superior image processing capabilities. While in retrieval and classification tasks, InternVL3 performs slightly

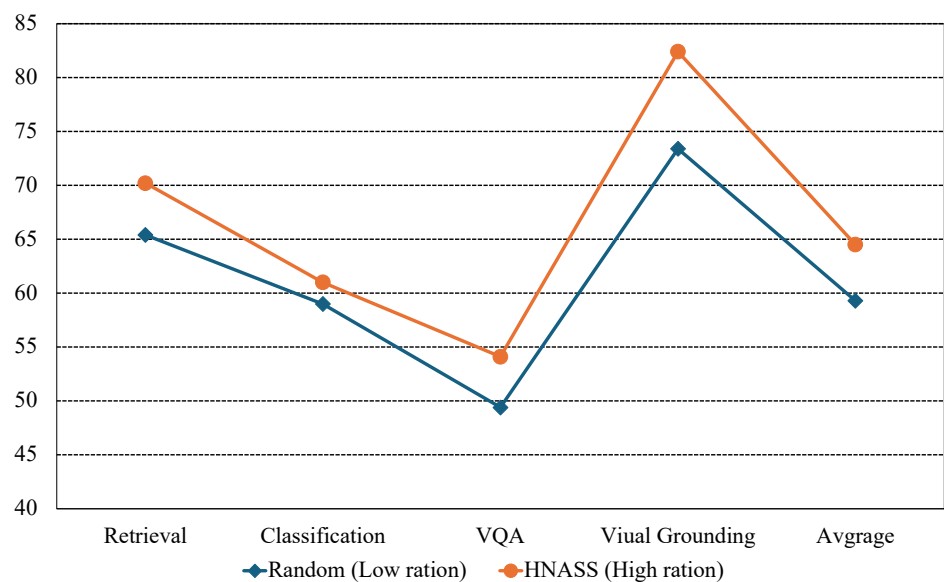

Figure 5: The effects of different ratios of hard negative samples on model performance.

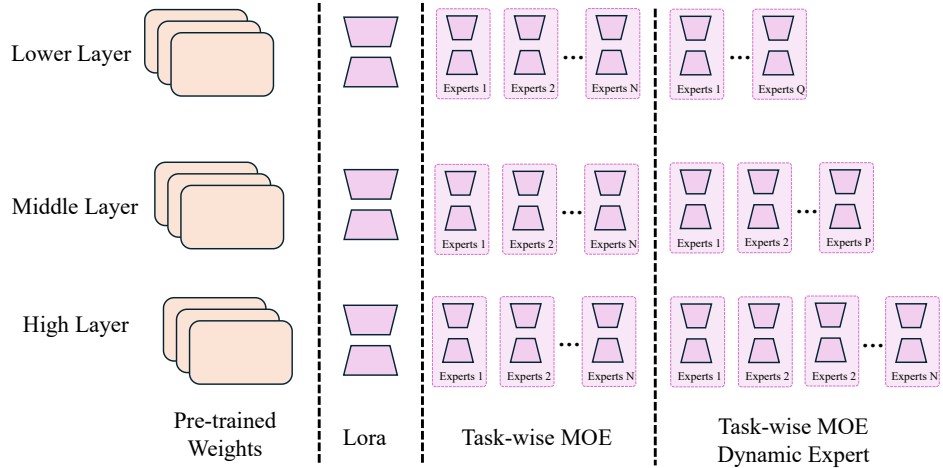

Figure 6: The architecture of M3E-Dynamic Expert.

worse than QWen-VL. From the comparison between the second and third rows, we can observe that regardless of the LVLMs backbone used, our method outperforms existing SOTA (state-of-the-art) methods and achieves the best performance. This further validates the superiority of our approach and demonstrates the adaptability of our framework.

### A.5.2 DIFFERENT LORA COMBINATIONS

In this section, we further explore different Task-wise MoE architecture designs. Previous research (Gao et al., 2024) has pointed out that when the number of experts is too large, the model will introduce redundancy due to representation collapse or overfitting of the routing strategy, thereby reducing overall performance. In particular, when modeling data at a more fine-grained level, introducing too many experts at a single layer can lead to overfitting of the representation to the training data, which in turn affects the model's generalization performance. Therefore, it is more reasonable to use different numbers of expert modules at different layers of the Transformer model.

Table 4: The effects of different Large Vision-Language Models backbone.

| Model | Retrieval | Classification | VQA | Viual Grounding | Avgrage |
|---|---|---|---|---|---|
| M$^3$E (Qwen2-VL-2B) | 72.1 | 65.9 | 64.8 | 86.4 | 69.9 |
| B3 (InternVL3-2B) | 69.0 | 62.6 | 64.0 | 86.9 | 67.8 |
| M$^3$E (InternVL3-2B) | 70.5 | 61.0 | 64.8 | 90.1 | 68.5 |

Table 5: The effects of different Task-wise MOE architectural designs, exploring how varying the number of task experts at each layer impacts performance.

| Model | Retrieval | Classification | VQA | Viual Grounding | Avgrage |
|---|---|---|---|---|---|
| VLM2Vec (Qwen2-VL-2B) | 65.4 | 59.0 | 49.4 | 73.4 | 59.3 |
| M$^3$E (Qwen2-VL-2B) | 72.1 | 65.9 | 64.8 | 86.4 | 69.9 |
| M$^3$E-Dynamic Expert (Qwen2-VL-2B) | 72.2 | 64.4 | 64.5 | 83.4 | 69.1 |

As shown in Fig. 6, based on this idea, we design a hierarchical expert allocation strategy for different layers of the model: only two task experts are introduced in the first eight layers to avoid excessive redundancy in processing token-level features at the bottom layer; four task experts are used in layers 8–16 to balance information richness and generalization during the gradual abstraction of mid-layers; and eight task experts are used in higher layers (after layer 16) to fully capture more abstract and complex high-level semantic features. Furthermore, at each layer, a routing mechanism is used to select the top two experts and shared experts to process the input.

The experimental results in Table 5 show that compared to the baseline method, our proposed dynamic expert allocation strategy can significantly improve the model's representational capabilities and downstream task performance. However, compared to the setting with a fixed number of experts, the effect of dynamic experts on the Visual Grounding task is slightly reduced. We speculate that this is because Visual Grounding relies more on the thorough modeling of underlying visual features, while our design uses a smaller number of experts at the low level, which may lead to a weakening of the ability to model the underlying image features, thus affecting task performance. Nevertheless, this result still verifies the effectiveness and superiority of dynamic expert allocation in the overall task and suggests that the expert allocation strategy can be further optimized in the low-level visual feature modeling in the future.

### A.5.3 THE IMPACT OF HARD NEGATIVE SAMPLES ON MODEL PERFORMANCE

As shown in Fig. 5, we selected different batch construction methods to observe the impact of varying ratios of hard negative samples on contrastive learning performance. The batches constructed randomly have a low proportion of hard negative samples, while batches constructed using HNASS contain a higher proportion of hard negative samples. We observe that as the proportion of hard negative samples increases, the performance of contrastive learning improves, demonstrating that an increase in hard negative samples enhances both the stability and effectiveness of the training.

Table 6: The detailed experimental results of several baselines and our $M^3E$ on MMEB-V1, which includes 20 in-distribution datasets and 16 out-of-distribution datasets. The 16 out-of-distribution datasets are highlighted with a yellow background in the table.

| | CLIP | OpenCLIP | SigLIP | BLIP2 | MagicLens | E5-V | UniIR | VLM2Vec | $M^3E$ |
|---|---|---|---|---|---|---|---|---|---|
| **Classification (10 tasks)** | | | | | | | | | |
| ImageNet-1K | 55.8 | 63.5 | 45.4 | 10.3 | 48.0 | 9.6 | 58.3 | 74.5 | 79.1 |
| N24News | 34.7 | 38.6 | 13.9 | 36.0 | 33.7 | 23.4 | 42.5 | 80.3 | 80.1 |
| HatefulMemes | 51.1 | 51.7 | 47.2 | 49.6 | 49.0 | 49.7 | 56.4 | 67.9 | 64.4 |
| VOC2007 | 50.7 | 52.4 | 64.3 | 52.1 | 51.6 | 49.9 | 66.2 | 91.5 | 82.3 |
| SUN397 | 43.4 | 68.8 | 39.6 | 34.5 | 57.0 | 33.1 | 63.2 | 75.8 | 81.8 |
| Place365 | 28.5 | 37.8 | 20.0 | 21.5 | 31.5 | 8.6 | 36.5 | 44.0 | 45.4 |
| ImageNet-A | 25.5 | 14.2 | 42.6 | 3.2 | 8.0 | 2.0 | 9.8 | 43.6 | 52.9 |
| ImageNet-R | 75.6 | 83.0 | 75.0 | 39.7 | 70.9 | 30.8 | 66.2 | 79.8 | 90.5 |
| ObjectNet | 43.4 | 51.4 | 40.3 | 20.6 | 31.6 | 7.5 | 32.2 | 39.6 | 52.0 |
| Country-211 | 19.2 | 16.8 | 14.2 | 2.5 | 6.2 | 3.1 | 11.3 | 14.7 | 30.2 |
| *All Classification* | 42.8 | 47.8 | 40.3 | 27.0 | 38.8 | 21.8 | 44.3 | 61.2 | 65.9 |
| **VQA (10 tasks)** | | | | | | | | | |
| OK-VQA | 7.5 | 11.5 | 2.4 | 8.7 | 12.7 | 8.9 | 25.4 | 69.0 | 65.7 |
| A-OKVQA | 3.8 | 3.3 | 1.5 | 3.2 | 2.9 | 5.9 | 8.8 | 54.4 | 60.8 |
| DocVQA | 4.0 | 5.3 | 4.2 | 2.6 | 3.0 | 1.7 | 6.2 | 52.0 | 92.5 |
| InfographicsVQA | 4.6 | 4.6 | 2.7 | 2.0 | 5.9 | 2.3 | 4.6 | 30.7 | 63.4 |
| ChartQA | 1.4 | 1.5 | 3.0 | 0.5 | 0.9 | 2.4 | 1.6 | 34.8 | 56.5 |
| Visual7W | 4.0 | 2.6 | 1.2 | 1.3 | 2.5 | 5.8 | 14.5 | 49.8 | 56.7 |
| ScienceQA | 9.4 | 10.2 | 7.9 | 6.8 | 5.2 | 3.6 | 12.8 | 42.1 | 47.7 |
| VizWiz | 8.2 | 6.6 | 2.3 | 4.0 | 1.7 | 2.6 | 24.3 | 43.0 | 52.0 |
| GQA | 41.3 | 52.5 | 57.5 | 9.7 | 43.5 | 7.8 | 48.8 | 61.2 | 69.4 |
| TextVQA | 7.0 | 10.9 | 1.0 | 3.3 | 4.6 | 8.2 | 15.1 | 62.0 | 82.8 |
| *All VQA* | 9.1 | 10.9 | 8.4 | 4.2 | 8.3 | 4.9 | 16.2 | 49.9 | 64.8 |
| **Retrieval (12 tasks)** | | | | | | | | | |
| VisDial | 30.7 | 25.4 | 21.5 | 18.0 | 24.8 | 9.2 | 42.2 | 80.9 | 83.9 |
| CIRR | 12.6 | 15.4 | 15.1 | 9.8 | 39.1 | 6.1 | 51.3 | 49.9 | 63.0 |
| VisualNews_t2i | 78.9 | 74.0 | 51.0 | 48.1 | 50.7 | 13.5 | 74.3 | 75.4 | 77.6 |
| VisualNews_i2t | 79.6 | 78.0 | 52.4 | 13.5 | 21.1 | 8.1 | 76.8 | 80.0 | 80.7 |
| MSCOCO_t2i | 59.5 | 63.6 | 58.3 | 53.7 | 54.1 | 20.7 | 68.5 | 75.7 | 78.0 |
| MSCOCO_i2t | 57.7 | 62.1 | 55.0 | 20.3 | 40.0 | 14.0 | 72.1 | 73.1 | 74.9 |
| NIGHTS | 60.4 | 66.1 | 62.9 | 56.5 | 58.1 | 4.2 | 66.2 | 65.5 | 66.9 |
| WebQA | 67.5 | 62.1 | 58.1 | 55.4 | 43.0 | 17.7 | 89.6 | 87.6 | 90.6 |
| FashionIQ | 11.4 | 13.8 | 20.1 | 9.3 | 11.2 | 2.8 | 40.2 | 16.2 | 25.6 |
| Wiki-SS-NQ | 55.0 | 44.6 | 55.1 | 28.7 | 18.7 | 8.6 | 12.2 | 60.2 | 66.6 |
| OVEN | 41.1 | 45.0 | 56.0 | 39.5 | 1.6 | 5.9 | 69.4 | 56.5 | 67.4 |
| EDIS | 81.0 | 77.5 | 23.6 | 54.4 | 62.6 | 26.8 | 79.2 | 87.8 | 90.5 |
| *All Retrieval* | 53.0 | 52.3 | 31.6 | 33.9 | 35.4 | 11.5 | 61.8 | 67.4 | 72.1 |
| **Visual Grounding (4 tasks)** | | | | | | | | | |
| MSCOCO | 33.8 | 34.5 | 46.4 | 28.9 | 22.1 | 10.8 | 46.6 | 80.6 | 76.3 |
| RefCOCO | 56.9 | 54.2 | 70.8 | 47.4 | 22.8 | 11.9 | 67.8 | 88.7 | 91.0 |
| RefCOCO-matching | 61.3 | 68.3 | 50.8 | 59.5 | 35.6 | 38.9 | 62.9 | 84.0 | 87.1 |
| Visual7W-pointing | 55.1 | 56.3 | 70.1 | 52.0 | 23.4 | 14.3 | 71.3 | 90.9 | 91.0 |
| *All Visual Grounding* | 51.8 | 53.3 | 59.5 | 47.0 | 26.0 | 19.0 | 65.3 | 86.1 | 86.4 |
| **Final Score (36 tasks)** | | | | | | | | | |
| All | 37.8 | 39.7 | 34.8 | 25.2 | 27.8 | 13.3 | 44.7 | 62.9 | 69.9 |
| All IND | 37.1 | 39.3 | 32.3 | 25.3 | 31.0 | 14.9 | 47.1 | 67.5 | 73.8 |
| All OOD | 38.7 | 40.2 | 38.0 | 25.1 | 23.7 | 11.5 | 41.7 | 57.1 | 63.1 |

