# OpenReview forum: "M3E: A Unified Framework for Large-Scale Multimodal Embedding via Multi-Task Mixture-of-Experts"
_ICLR.cc/2026/Conference — ICLR 2026 Conference Withdrawn Submission_

### Official Review · Reviewer_bUtV · 2025-10-28

**Soundness:** 3
**Presentation:** 3
**Contribution:** 2
**Rating:** 4
**Confidence:** 4

**Summary:**

This paper proposes M3E, a unified multimodal multi-task embedding framework that leverages Large Vision-Language Models (LVLMs) to learn universal cross-task representations. At the data level, a Hard Negative-Aware Sample Scheduler (HNASS) is designed: a teacher model constructs a similarity graph, followed by graph clustering to raise the ratio of hard negatives and filter easy/false negatives in each batch. At the model level, Task-wise Low-Rank Mixture-of-Experts (Task-wise MOE) splits parameters into task-specific and shared experts, dynamically combined by a router to mitigate task conflicts and forgetting. Joint training on 36 datasets across four task types significantly surpasses prior work, improving average scores by 10.3 and 7.8 points for 2B- and 7B-parameter LVLMs, respectively.

**Strengths:**

1. The work is well-motivated.  HNASS and Task-wise MOE are designed for tackling easy samples and multi-task challenges respectively.
2. Compared to some baseline models, the model achieves the state-of-the-art on several different tasks on different LVLM backbones.

**Weaknesses:**

1. HNASS relies on an iteratively updated teacher model, complicating training and raising computational cost. Hyper-parameters p, m, c need manual tuning, transferability across datasets/scales is unclear.
2. Number and layer allocation of task experts are empirical. Downstream tasks are uniformly cast as ranking; generative or more complex scenarios are not covered. Routing weights and expert specialization lack quantitative analysis, hampering failure diagnosis.

**Questions:**

1. How about the sensitivity of p and m in HNASS across datasets?
2. How about the impact of expert initialization, routing noise, and regularization on convergence in Task-wise MOE?
3. How about extra training/inference time and memory caused by HNASS clustering and MOE routing?

---

### Official Review · Reviewer_wbgn · 2025-10-31

**Soundness:** 3
**Presentation:** 3
**Contribution:** 3
**Rating:** 4
**Confidence:** 3

**Summary:**

This paper introduces M3E, a multimodal multi-task embedding framework that jointly enhances data sampling and model architecture. On the data side, the Hard Negative-Aware Sample Scheduler (HNASS) builds similarity graphs from embeddings to mine hard negatives via graph clustering. On the model side, the Task-wise Low-Rank Mixture-of-Experts (Task-wise MoE) uses task-specific and shared LoRA experts with a routing mechanism to reduce gradient conflict and task forgetting. Experiments on MMEB-V1 (36 tasks) show consistent improvements over strong LVLM baselines.

**Strengths:**

1. Innovative data-model synergy: HNASS complements Task-wise MoE to address both false negatives and multi-task interference.
2. Well-motivated low-rank MoE design that balances efficiency and specialization.
3.Strong quantitative results with +10.3 and +7.8 point gains on Qwen2-VL-2B/7B baselines.

**Weaknesses:**

1. Limited analysis of computational overhead (e.g., graph clustering, expert routing latency).
2. Ablations on dynamic expert allocation show mixed effects (e.g., lower Visual Grounding).
3. Lack of comparison with alternative negative-sampling or multi-task scheduling methods. (I am not sure whether there are many negative-sampling approaches.)

**Questions:**

1. How does M3E scale computationally compared to random batching or B3++?
2. Can the teacher update schedule or similarity thresholds (p,m) be tuned automatically?
3. What is the runtime/memory cost at inference when multiple experts are active?
4. Why does dynamic-expert allocation harm Visual Grounding, and could low-layer experts mitigate this?

**Details Of Ethics Concerns:**

No.

---

### Official Review · Reviewer_yMWm · 2025-10-31

**Soundness:** 2
**Presentation:** 3
**Contribution:** 2
**Rating:** 2
**Confidence:** 4

**Summary:**

This paper focuses on two problems in the subject of multi-modal embedding representation: 1) hard negative sample mining; and 2) task conflict and oblivion. The problems are well described, but there lacks of novel strategies and new valid metrics to measure the effectiveness on these problems (especially on the task conflict and oblivion).

**Strengths:**

- The research problems are focused. The negative sample mining is an important subject in contrastive learning, while task conflict & oblivion are inspiring.
- M3E surpasses other baselines and reaches good performance.

**Weaknesses:**

- The overall methodology is based on B3++ and lacks of novelty. M3E shares the same strategy with B3++ which focuses on batch mining with ranking and METIS clustering. The hard negative sample mining problem is not new.
    - Compared with B3++, the iterative teacher model updating is not clear. How do you refine the teacher model and the representations accordingly to calculate similarities (line 301 - 302)?
- The task conflict & oblivion problems are not well formed. Although the concept is clearly stated in the introduction, the preliminary experiments (in Figure 2) do not fully represent the phenomenon.
    - The meaning of Figure 2b x-axis is unclear, is it in an early stage of training or the model has been well trained? It is not clear to confirm whether it is performance fluctuation or real task conflict & oblivion.
    - What if there are only two tasks? If there are conflicts, the phenomenon would be the same? One obtains higher performance during training, and the other one will degrade.
    - Although the overall performance is improved, there lacks metrics to measure the levels of task conflict and oblivion. How much do you solve them?
- Additional experts means additional computational costs. Although the performances are improved, it is not clear if it comes from these additional trainable parameters.
    - Besides, more shared experts lead to higher performance, but the top-k strategy is worse than the mask-based strategy. Maybe an additional ablation experiment on fully activated experts / scaling the number of top-k experts would help understanding the dynamics?

**Questions:**

- Figure 1: Archo should be Anchor?
- MOE → MoE
- B/C → B/c in line 291
- Could you provide the whole histograms with thresholds in them rather than presenting the numbers of three categories in Figure 2a?

---

### Note · Authors · 2025-11-23

I have read and agree with the venue's withdrawal policy on behalf of myself and my co-authors.